

# Design of compensation algorithms for zero padding and its application to a patch based deep neural network

Safi Ullah[1,2] and Seong-Ho Song[1]

[1] Division of Software, Hallym University, Chuncheon, Gangwon-do, Republic of Korea
[2] Department of Information Technology, University of Gujrat, Gujrat, Punjab, Pakistan

## ABSTRACT

In this article, compensation algorithms for zero padding are suggested to enhance the performance of deep convolutional neural networks. By considering the characteristics of convolving filters, the proposed methods efficiently compensate convolutional output errors due to zero padded inputs in a convolutional neural network. Primarily the algorithms are developed for patch based SRResNet for Single Image Super Resolution and the performance comparison is carried out using the SRResNet model but due to generalized nature of the padding algorithms its efficacy is tested in U-Net for Lung CT Image Segmentation. The proposed algorithms show better performance than the existing algorithm called partial convolution based padding (PCP), developed recently.

## INTRODUCTION

In convolutional neural networks (*Shinde & Shah, 2018*), there are situations such as skip-connections (*He et al., 2016b*) in residual networks (*He et al., 2016a*) where padding extra pixels to an input is required before convolution to obtain an output of the same size as an input. The amount of required padding depend on the size of convolving filters and striding value. Padding method particularly becomes more important in the case of smaller (patch) inputs, where the ratio of padded pixels to the input pixels gets higher.

There are many perspectives of deep learning models which are being addressed by the research community *e.g.*, Network architectures (*He et al., 2016a*; *Huang et al., 2017*), Network initialization (*Ji et al., 2023*; *Solanki & Shah, 2023*), Network optimization (*Chung & Sohn, 2023*; *Ioffe & Szegedy, 2015*), Activation functions (*Dubey, Singh & Chaudhuri, 2022*; *Nwankpa et al., 2018*), Network pruning (*Chang et al., 2023*; *Poyatos et al., 2023*), *etc.* but padding schemes are not paid much attention even though they can affect the performance of deep neural networks. Simple padding algorithms such as zero padding and reflection padding have been applied to neural networks for convenience, which can lead to erroneous performance like potential gray areas developed by deep learning models

Corresponding author
Seong-Ho Song, ssh@hallym.ac.kr

(*Alsallakh et al., 2020*). A detailed comparison of various padding schemes is made in *Alsallakh et al. (2020)*.

In zero padding, zeros are simply padded to the boundaries of the input (*Krizhevsky, Sutskever & Hinton, 2017*). It is simple and easy to implement. In many computer vision applications like classification (*Gao et al., 2019*; *Wightman, Touvron & Jégou, 2021*; *Xu et al., 2022*) and segmentation (*Chen et al., 2018*; *Khan et al., 2021*; *Yu & Koltun, 2015*; *Zhu et al., 2019*), it is known that deep neural network models can easily adapt to the padded zeros. By the way the performance of deep neural network models, where skip connections are required, can be deteriorated because it is more sensitive to the zero padding algorithm. Replication padding replicates the border pixels to extend the input size (*Cheng et al., 2018*). Contrary to zero padding, replication padding uses the input data itself to generate padding data. Replication padding (symmetric padding) is useful in cases where the border area is not varying much and does not contain many details. In reflection padding (or mirroring), the pixels near the border are repeated first. Reflection padding possesses properties that are opposite to replication padding, *i.e.,* this method is useful when the border area has more details as compared to the non-border area. Both the replication and reflection methods stretch the border pixels, consequently the original distribution of the input data is altered (*Nguyen et al., 2019*). In linear extrapolation based padding the padded pixels are extrapolated from the input pixels using the linear extrapolation method. Models trained with linear extrapolated padding are too sensitive to the border pixels. A more systematic approach is proposed by *Innamorati et al. (2018)*, to introduce filters to layers in a neural network to learn padding pixels, which loads additional burden to train extra filters for simple padding. Cube padding by *Cheng et al. (2018)* is suggested based on the image projection method to deal with boundaries, where the image is first projected on a cube and the cube faces are concatenated to construct a 2D image. In distribution padding (*Nguyen et al., 2019*), a local mean of a window sliding over a border region is calculated and padded, so the local distribution is maintained unlike zero padding, replication padding, and reflection padding. A separate module for padding have proposed by *Alrasheedi, Zhong & Huang (2023)* which learn the padded pixels from the borders of the input by stacking the borders and constructing a 2D matrix $M$ as an input to the predictor. Padding modules are placed at the desired position in a deep learning model to pad extra pixels. Padding module resolves the issue of non-trainable methods but it has a major disadvantage that the shape of an input should be square, *i.e.,* the width and height of the input must be same so that the stacking matrix $M$ can be constructed.

Recently, partial convolution based padding (PCP) was introduced by *Liu et al. (2018)* and showed good performance in many applications. It can efficiently handle problem of boundary artifact by introducing a compensation ratio which is just multiplied to the convolution result obtained using zero padding. Even though it is very simple and efficient in some cases, the PCP has intrinsic drawback that it can work appropriately only in limited situations where the values of both filter's elements and convolutional outputs are nonnegative. Depending on the type of an activation function for layer outputs in a neural network, the layer outputs can be negative. Furthermore, parameter values of

the convolving filters should not be regulated to be nonnegative in general during neural network learning.

In this article, the original PCP algorithm is revisited and modified versions of PCP algorithms called signed PCP and adaptive PCP are suggested to consider more general situations where filter's elements in a neural network are not assumed to be nonnegative and enhance the performance of the original PCP.

## PARTIAL CONVOLUTION BASED PADDING REVISITED

Convolution operation with zero padded inputs results in misleading outputs while the need to maintain the same sizes between inputs and outputs in a network is satisfied by zero padding (*Alsallakh et al., 2020*). As convolution operation with zero padded inputs continues through the layers in a neural network, this kind of phenomenon becomes more severe as layers go deeper. For example, in case of patch input based single image super resolution (*Ullah & Song, 2023*), due to simple zero padding on the boundary pixels of each patch input, the boundary pixels in the super-resolution output patches are not properly generated and the distortions are more clearly visible in generated super-resolution images. Padding algorithms should be carefully chosen in order to effectively handle boundary effects due to padded zeros.

Partial convolution based padding (PCP) (*Liu et al., 2018*) is suggested to compensate convolution errors caused by zero padding. In partial convolution based padding, a compensation ratio is introduced to compensate distortions due to zero padding and PCP algorithm is trying to correct convolutional outputs by multiplying it to a convolutional output with zero padding according to Eq. (1).

$$y_{i,j} = r_{i,j} \bigodot \hat{y}_{i,j} \tag{1}$$

where $\hat{y}_{i,j}$ and $y_{i,j}$ are respectively an output after filter convolution with zero padded inputs and a compensated output after PCP algorithm is applied. In Eq. (1), $r_{i,j}$ is a compensation ratio for the pixel position $(i,j)$ and is defined by Eq. (2).

$$r_{i,j} = \frac{n_{i,j} + c_{i,j}}{n_{i,j}} \tag{2}$$

where $n_{i,j}$ is the number of filter's elements overlapped with actual input pixels, and $c_{i,j}$ is the number of filter's elements overlapped with padded pixels during convolution (*Liu et al., 2018*). Note that $n_{i,j}$ and $c_{i,j}$ in Eq. (2) are always nonnegative. The compensation ratio $r_{i,j}$ is always larger than 1 for a pixel where convolving filter overlaps with padded pixels at the boundary, that is $c_{i,j} > 0$. The performance superiority of PCP over zero padding algorithm has been investigated in *Liu et al. (2018)*, *Liu et al. (2022)* and *Ullah & Song (2023)*.

In PCP, output of convolution operation is compensated by multiplying a ratio given by Eq. (2) to a convolutional output using zero padding and compensated output is always amplified since compensation ratios around boundary pixels are always larger than 1. By the way, filter's elements in a deep learning model can be negative in the course of learning. When some of filter's elements are negative and the trimmed pixels of the input (which are

replaced by padded zeros) are positive, compensated convolutional outputs should become smaller than convolutional outputs using zero padding but PCP generates amplified results. So the performance of PCP can be improved by considering the characteristics of the filter's elements which are overlapped with padded zeros.

## MODIFIED COMPENSATION ALGORITHMS OF PCP

As discussed in the previous section, the original PCP algorithm takes into account only the number of filter's elements overlapped with actual input pixels and the number of filter's elements overlapped with padded pixels when it calculates a compensation ratio $r_{ij}$. So it may lead to a wrong estimation for compensated convolution results in some situations by simply considering the number of filter's elements overlapped with actual input and padded pixels. The performance of PCP can be improved if parameters of the convolving filters are considered rather than only the number of filter's elements overlapped with actual input and padded pixels.

Consider an example illustrated in Fig. 1. Some of the filter's elements are negative and all the pixel values of the input are positive. Let's consider the convolution operation with a 6×6 input shown in Fig. 1A and 3×3 filter shown in Fig. 1B. 3×3 inputs depicted in (D) and (E) of Fig. 1 are respectively obtained by padding original boundary pixel values and zero values to 2×2 input shown in Fig. 1C which is extracted from the upper left part of the upper 6×6 input depicted in Fig. 1A. If we consider the calculation of convolutional output corresponding to the pixel of the value, 5, the top left pixel in (C), the true value of convolutional output corresponding to the pixel, 5 is obtained using the 3×3 input (d) and is −11. In case of zero padding, the convolution is calculated using the zero padded 3×3 input (e) and the result is 7. In this example, $n_{ij}$ is 4 and $c_{i,j}$ is 5. Then, $r_{i,j}$ is $\frac{9}{4}$ by Eq. (3). Using PCP given in Eq. (1) with $\hat{y}_{i,j} = 7$, the compensated output $y_{i,j}$ is 15.75. So the PCP algorithm amplifies the error rather than reducing the value because zero padding result is 7 and true convolutional output value is −11. This results from the fact that some of the filter's elements are negative.

In this article, two modified versions of PCP called signed PCP and adaptive PCP are suggested based on the fact that the values of the filter's elements are changed during the course of learning. Hence considering the characteristics of a convolving filter's elements instead of simply using the number of filter's elements, as in PCP, can improve the performance of PCP.

### Signed PCP

The original PCP algorithm works in a proper way when the parameters of a convolving filter are assumed to be nonnegative, which is not always possible in the period of deep learning model's training. Throughout this article, it is assumed that ReLU activation function is used in each layer, which means that the output of each layer in a deep learning model is nonnegative.

In order to consider filter's characteristics in calculating the compensation ratio $r_{i,j}$, let's redefine a parameter $c_{i,j}$ in Eq. (2) as the sum of signs of filter's elements which are overlapped with padded pixels. Let's consider the example in Fig. 1 again. Then, the

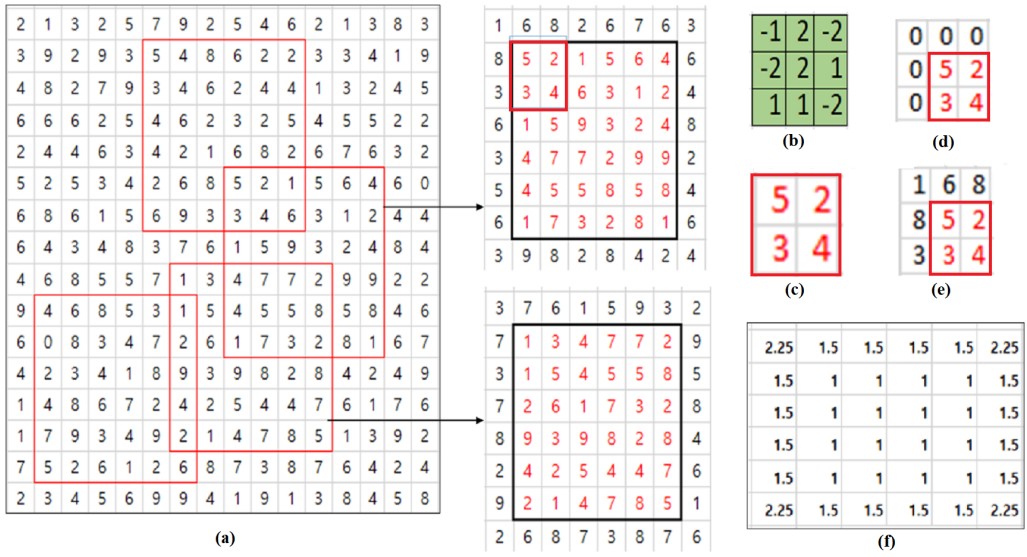

**Figure 1   PCP Example.** (A) Input patches with different contents. (B) Filter. (C) 2×2 input. (D) Zero padded input. (E) Input padded with original values. (F) Ratio matrix of PCP for 6×6 input.

value of a parameter $c_{i,j}$ in Eq. (2) is the sum of $1, -1, -1, 1, -1$ which are the signs of filter's elements overlapped with the padded pixels. So $c_{i,j}$ in Eq. (2) is equal to $-1$ and the compensation ratio $r_{i,j}$ is $\frac{3}{4}$ with the same $n_{i,j} = 4$. So the compensated value $y_{i,j}$ of the zero padded convolutional output $\hat{y}_{i,j} = 7$ is calculated to be 5.25 which is closer to the true convolutional output value, $-11$ than the value 15.75 calculated by the original PCP algorithm.

So the compensation performance of PCP algorithm can be improved by considering the signs of filter's elements which are overlapped with padded pixels. A modified version of the original PCP algorithm called as signed PCP (sPCP) is defined as follows.

$$y_{i,j}^s = r_{i,j}^s \bigodot \hat{y}_{i,j} \tag{3}$$

where $y_{i,j}^s$ and $\hat{y}_{i,j}$ are respectively a compensated output after signed PCP and an output after convolution with zero padded inputs, and $r_{i,j}^s$ is a compensation ratio for the pixel position $(i,j)$ in case of signed PCP. In Eq. (3), the compensation ratio $r_{i,j}^s$ is defined by Eq. (4).

$$r_{i,j}^s = \frac{n_{i,j} + c_{i,j}^s}{n_{i,j}} \tag{4}$$

where the definition of $n_{i,j}$ is the same as in Eq. (2) and $c_{i,j}^s$ is the sum of signs of filter's elements overlapped with padded pixels.

A pseudo code of a skip connection block with Signed PCP is given in Algorithm 1.

---

**Algorithm 1: Skip connection block with Signed PCP layer**

**Input**: Tensor $N$, filter $K$, Conv2d, $stride_x$, $stride_y$

**Output**: Tensor $N'$

**Step 1**: get the output from *conv2d* layer using zero padding

$\hat{y} = \text{conv2d}(\text{Tensor } N, \text{filter } K, (stride_x, stride_y), padding = 'same')$

**Step 2**: compensate the zero padded output $\hat{y}_{i,j}$ with compensation ratio $r_{i,j}^s$

calculate compensating ratio matrix $r^s$ as

$r_{i,j}^s = \frac{n_{i,j} + c_{i,j}^s}{n_{i,j}}$

get the Hadamard product of $r_{i,j}^s$ and $\hat{y}_{i,j}$ to get the compensated output $y^s$

$y_{i,j}^s = r_{i,j}^s \odot \hat{y}_{i,j}$

**Step 3**: get $N'$ as

$N' = N + y^s$

**end**

---

## Adaptive PCP

sPCP can improve the performance of the original PCP algorithm by considering the signs of filter's elements but it is still inefficient because it does not concern the exact values of filter's elements.

In calculating the compensation ratio $r_{i,j}$, let us consider exact values of filter elements in calculating a parameter $c_{i,j}$ in Eq. (2). Instead of the sum of signs of filter's elements which are overlapped with padded pixels, the parameter $c_{i,j}$ in Eq. (2) is obtained using the sum of exact values of filter's elements. Let's consider the case in Fig. 1 again. Then, the value of a parameter $c_{i,j}$ in Eq. (2) is the sum of $\{1, -2, -1, 2, -2\}$ which are the values of filter's elements overlapped with padded pixels. So $c_{i,j}$ in Eq. (2) is equal to $-2$ and the compensation ratio $r_{i,j}$ is $\frac{2}{4}$ with the same $n_{i,j} = 4$. So the compensated value $y_{i,j}$ of the zero padded convolutional output $\hat{y}_{i,j} = 7$ is calculated to be 3.5 which is closer to the true convolutional output value, $-11$ than the value 5.25 calculated by the signed PCP algorithm. While sPCP and PCP consider the contribution of each filter's element to the convolution operation evenly, the convolution error can be improved by considering exact values of filter's elements.

Another new modified PCP algorithm called adaptive PCP (aPCP) is defined as follows.

$$y_{i,j}^a = r_{i,j}^a \odot \hat{y}_{i,j} \tag{5}$$

where $y_{i,j}^a$ and $\hat{y}_{i,j}$ are respectively a compensated output after adaptive PCP and an output after convolution with zero padded inputs, and $r_{i,j}^a$ is a compensation ratio for the pixel position $(i,j)$ in case of adaptive PCP. In Eq. (5), the compensation ratio $r_{i,j}^a$ is defined by Eq. (6).

$$r_{i,j}^a = \frac{n_{i,j} + c_{i,j}^a}{n_{i,,j}} \tag{6}$$

where the definition of $n_{ij}$ is the same as in Eq. (2) and $c_{i,j}^a$ is the sum of the exact values of filter's elements overlapped with padded pixels.

A pseudo code of a skip connection block with Adaptive PCP is given in Algorithm 2.

---

**Algorithm 2: Skip connection block with Adaptive PCP layer**

---

**Input**: Tensor $N$, filter $K$, Conv2d, $stride_x$, $stride_y$

**Output**: Tensor $N'$

**Step 1:** get the output from *conv2d* layer using zero padding

$\hat{y} = \text{conv2d}(\text{Tensor } N, \text{filter } K, (stride_x, stride_y), padding = 'same')$

**Step 2:** compensate the zero padded output $\hat{y}_{i,j}$ with compensation ratio $r_{i,j}^a$

calculate compensating ratio matrix $r^a$ as

$r_{i,j}^a = \frac{n_{i,j} + c_{i,j}^a}{n_{i,j}}$

get the Hadamard product of $r_{i,j}^a$ and $\hat{y}_{i,j}$ to get the compensated output $y^a$

$y_{i,j}^a = r_{i,j}^a \odot \hat{y}_{i,j}$

**Step 3:** get $N'$ as

$N' = N + y^a$

**end**

---

Remark 1: Note that the compensation ratios for pixel positions which are not adjacent to padded pixels are always 1 in all three techniques. That means all the PCP algorithms don't have to update ratios and compensate the convolutional outputs with zero padding for inside pixels. In the original PCP, the compensation ratios can be calculated before training and fixed once they are calculated. By the way, sPCP and aPCP require additional computation time before each iteration to update ratio values once the values of the filter's elements are updated by backpropagation, but it does not take too much time because only the compensation ratios of the boundary pixels need to be updated

Remark 2: Compared to the zero padding case, all three techniques need additional memory spaces for compensation ratios. Table 1 summarizes computational resource requirements in case of SRResNet introduced in *Ullah & Song (2023)*. Considering compensation ratios for every element of feature maps, the number of parameters of PCP cases are just doubled with respect to zero padding case but it can be considerably reduced if the update of compensation ratios only for boundary pixels is considered because the compensation ratios for inside pixels are always fixed to be 1.

In the next section, performance analysis will be investigated using patch input based SRResNet (*Ullah & Song, 2023*) where padding algorithms are necessary because of skip connections in the deep neural networks and have severe influence on the network performance

| Model | Parameters (Million) | FLOPs (Billion) | GPU Memory Required (GB) |
|---|---|---|---|
| SRResNet (input size 192×192) | 1.7221 | 1.4023 | 8.3004 |
| SRResNetp (input size 16×16) | 1.7221 | 0.0097325 | 0.0592 |
| SRResNet-pcp | 1.7230 | 0.0097892 | 0.0608 |
| SRResNet-spcp | 1.7238 | 0.0098207 | 0.0814 |
| SRResNet-apcp | 1.7238 | 0.0195784 | 0.0814 |

**Table 1  Computational resources.**

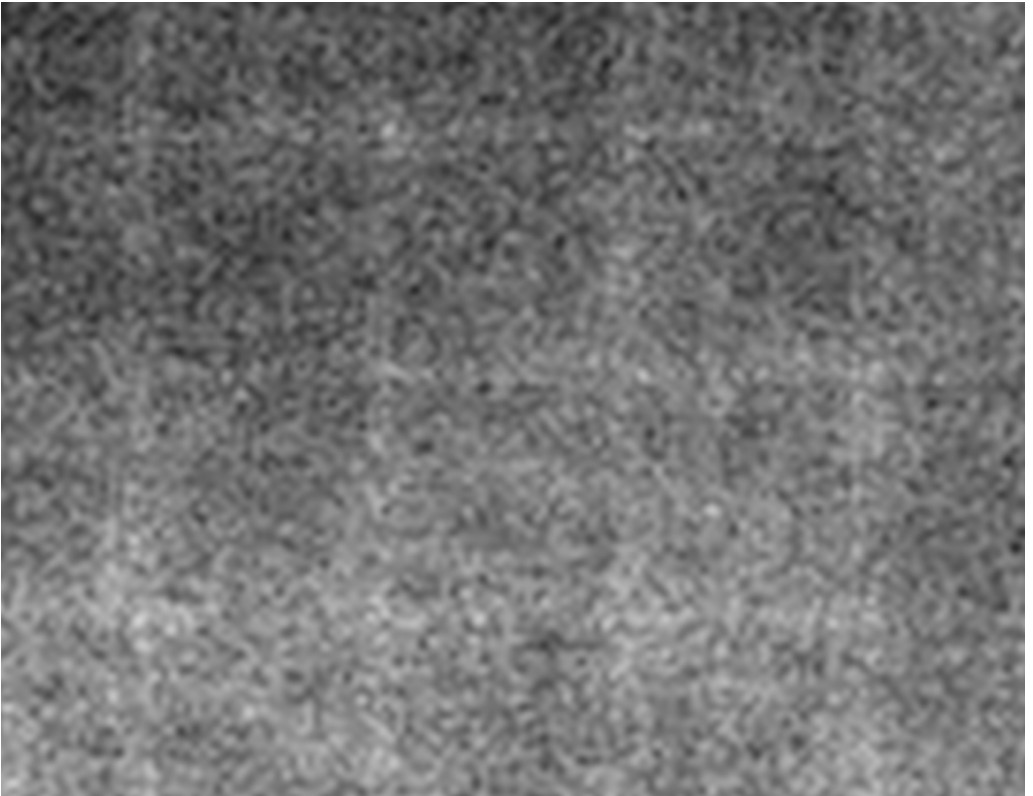

**Figure 2   Difference image between reconstructed SR and target HR image.**

## PERFORMANCE ANALYSIS: PATCH BASED SRRESNET

In this section, performance analysis of the proposed algorithms is carried out using a patch based SRResNet. In a patch based super-resolution(SR) deep neural network, it is relatively more critical to compensate the zero padding errors for performance enhancement as shown in Fig. 2. The figure illustrates the difference image between the original image and the super-resolution image obtained from the patch based super-resolution residual neural network called SRResNet (*Ullah & Song, 2023*) where zero padding is used.

SRResNet is a customized ResNet for image super-resolution and has residual blocks with skip connections which can manage vanishing gradient issues effectively in deep neural

networks. As mentioned in the previous section, an appropriate padding algorithm should be adopted to implement residual blocks with skip connections in SRResNet, which can influence on the performance of SRResNet. So SRResNet is a good candidate of deep learning neural networks for the performance comparison of padding algorithms.

SRResNet architecture of *Ullah & Song (2023)* is used for super resolution task. The model works with four residual blocks where $conv2d - pcp$ is a convolutional layer with each convolving filter of size $k$ and $n$ is the total number of convolutional filters. As an input, a small patch image is applied to SRResNet for super-resolution, which can facilitate the training of complex networks more efficiently despite limited computational capability and provide comparably good performance (*Ullah & Song, 2023*). As mentioned before, padding algorithms are inevitably necessary to implement residual blocks with skip connections in ResNet and zero padding is simply utilized in conventional ResNet. Padding algorithms should be carefully selected for a patch image based SRResNet because they can affect the performance considerably.

In order to analyze the performance of padding algorithms for SRResNet, a total of 40 images of dimension $640 \times 640 \times 3$ from COCO 2017 (*Lin et al., 2014*) are randomly selected for the task of 2x super-resolution of input images. Among them, 36 images are selected to train SRResNet using each padding algorithm (zero padding, PCP, sPCP, aPCP) and four images are chosen for validation. The trained SRResNet model is tested on three well-known benchmark datasets such as Set5 (*Bevilacqua et al., 2012*) with five images, Set14 (*Zeyde, Elad & Protter, 2010*) with 14 images, and BSD100 (*Martin et al., 2001*) with 100 images.

To implement a deep learning SRResNet with each padding algorithm for performance analysis, Python 3.6, Tensorflow-gpu version 1.14 and Keras 2.3 are adopted as a software framework on the hardware platform with an NVIDIA GPU RTX2070 (8GB GDDR6). In order to train a SRResNet using patch image inputs, a total number of patches of size $32 \times 32 \times 3$ are extracted from 36 images of size $640 \times 640 \times 3$.

Figure 3 illustrates the images consisting of average values of pixel differences between sample original HR images and corresponding 2x super-resolution ones for each padding algorithm. Here average pixel differences are calculated pixel-wise for 77 of sample original HR images from BSD100 test set. As seen in Fig. 3, the boundaries are more clearly observed in zero padding case, which means that zero padding algorithm causes more errors in the boundary pixels than all the PCP algorithms. It can be seen from Fig. 3D that adaptive PCP can compensate the errors in the boundary areas more efficiently than the other algorithms.

In Table 2, the effectiveness of PCP algorithms over zero padding is numerically described. The performance of each algorithm is summarized separately for the inside area and the padded boundary area of each images in Fig. 3. All PCP algorithms result in smaller mean square errors (MSE) in both inside and padded boundary areas than zero padding algorithm. It can be seen that MSE difference between PCP's and zero padding in the padded area is larger than the one of the inside area, which means that all PCP's are effectively compensating zero padding effects in the boundary padded area as expected. As shown in the table, aPCP shows better results than other PCP's.

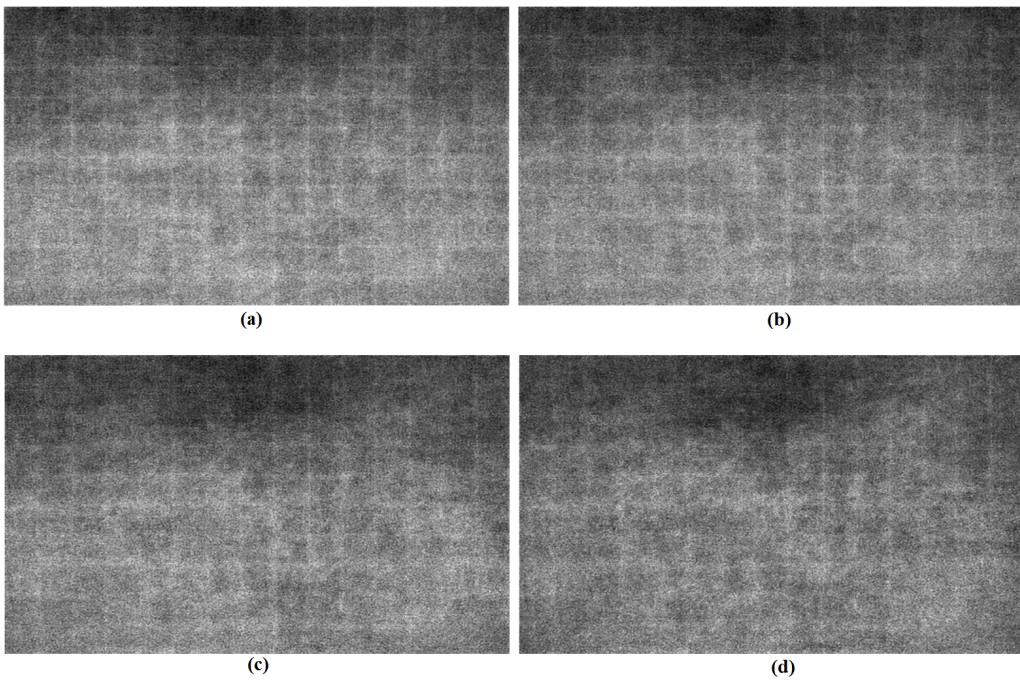

**Figure 3 Assembled images of pixel errors between super-resolution and high-resolution images.** (A) Zero padding. (B) Partial convolution based padding. (C) Signed PCP. (D) Adaptive PCP.

**Table 2 Performance analysis of padding algorithms.**

| Average MSE | Zero padding (ZP) | PCP | Difference (ZP-PCP) | sPCP | Difference (ZP-sPCP) | aPCP | Difference (ZP-aPCP) |
|---|---|---|---|---|---|---|---|
| MSE of Inner part | 86.227 | 83.384 | 2.84 | 76.906 | 9.32 | 73.256 | 12.97 |
| MSE of Padded part | 76.310 | 72.632 | 3.68 | 66.440 | 9.87 | 63.230 | 13.08 |

Table 1 summarizes computational resources needed for each SRResNet with a different padding algorithm. In Table 1, SRResNet and SRResNetp mean respectively the original SRResNet with the input image size of 192×192 and the one with 16×16 patch input images. The original SRResNet adopts zero padding algorithm. SRResNet-pcp, SRResNet-spcp and SRResNet-apcp in the table represent the SRResNetp's using respectively original PCP, signed PCP (sPCP) and adaptive PCP (aPCP) as a padding algorithm. As predicted, SRResNetp using aPCP needs more resources than other algorithms because of its increased complexity for the calculation of compensation ratios explained in Section 3. While the value of FLOPS of SRResNet-apcp is almost twice larger than the ones in case of the original PCP and the signed PCP due to its increased complexity, the performance is improved almost 1.4 times in terms of MSE (mean square error) than the original PCP and the signed PCP for the marked area shown in Fig. 4.

The performance of padding algorithms is evaluated in various aspects using various performance measures and summarized in Table 3. In Table 3, MSE and Peak Signal to

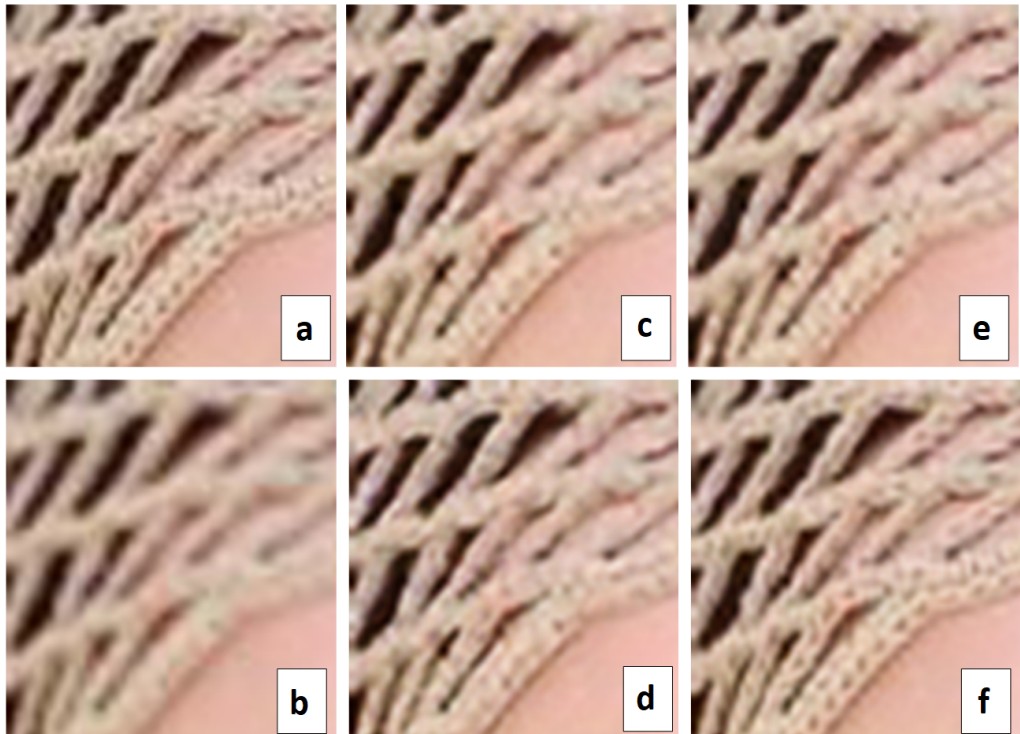

**Figure 4** **Visual comparison.** (A) Original HR image, (B) Bicubic interpolation (MSE: 245.815) (C) SR image using zero-padding (MSE: 197.118), (D) SR image using PCP(MSE: 168.955), (E) SR image using sPCP (MSE: 163.913), (F) SR image using aPCP (MSE: 120.089).

Noise Ratio (PSNR) are the most frequently adopted as performance measures in pixel-level performance evaluations. However, these two measures are insufficient for image similarity evaluation in some cases. So various performance measures for the evaluation of image level similarity such as SSIM (Structural Similarity Index Measure (*Zhou, 2004*)), FSIM (Feature based Similarity Index Measure (*Zhang et al., 2011*)), UIQ (Universal Image Quality index (*Wang & Bovik, 2002*)), ISSM (Information theoretic-based Statistic Similarity Measure (*Aljanabi et al., 2019*)) are also adopted as performance measures in Table 3. For all these image level similarity measures, higher value means the better performance. As seen in Table 3, adaptive PCP, aPCP shows the best performance in all the measures because it considers both the sign and magnitude of filter's elements. The performance of sPCP is also better than the original PCP as expected since it considers signs of filter's elements rather than only considering the number of padded pixels.

In Fig. 4 the performance of each algorithm is visually compared using a sample image. In Fig. 4A is the enlarged image of the marked area in the original HR sample image, Fig. 4B is the Bicubic interpolated image corresponding to the marked area, Fig. 4C is the super-resolved image using zero-padding, Fig. 4D is the super resolved image using partial convolution based padding, Fig. 4E is the super-resolved image using signed PCP, and Fig. 4F is the super-resolved image using adaptive PCP. In the figure, MSE value of

**Table 3  Performance comparison of padding algorithms in SRResNet.**

| Model | Set | MSE | FSIM | SSIM | UIQ | SRE | ISSM |
|---|---|---|---|---|---|---|---|
| Zero padding | Set5 | 39.513 | 0.794 | 0.922 | 0.825 | 57.350 | 0.750 |
| | Set14 | 101.850 | 0.774 | 0.866 | 0.757 | 58.198 | 0.690 |
| | BSD100 | 115.657 | 0.763 | 0.864 | 0.772 | 56.406 | 0.649 |
| | Average | 110.833 | 0.765 | 0.866 | 0.772 | 56.657 | 0.658 |
| pcp | Set5 | 38.616 | 0.797 | 0.924 | 0.828 | 57.397 | 0.751 |
| | Set14 | 97.577 | 0.777 | 0.867 | 0.759 | 58.330 | 0.691 |
| | BSD100 | 110.615 | 0.765 | 0.866 | 0.773 | 56.469 | 0.649 |
| | Average | 106.055 | 0.767 | 0.868 | 0.774 | 56.727 | 0.658 |
| sPCP | Set5 | 39.217 | 0.799 | 0.925 | 0.831 | 57.401 | 0.753 |
| | Set14 | 97.271 | 0.779 | 0.861 | 0.761 | 58.382 | 0.694 |
| | BSD100 | 109.819 | 0.771 | 0.868 | 0.778 | 56.497 | 0.650 |
| | Average | 105.379 | 0.773 | 0.869 | 0.778 | 56.757 | 0.659 |
| aPCP | Set5 | 36.040 | 0.801 | 0.927 | 0.836 | 57.541 | 0.757 |
| | Set14 | 96.659 | 0.781 | 0.875 | 0.772 | 58.379 | 0.701 |
| | BSD100 | 107.546 | 0.776 | 0.870 | 0.779 | 56.552 | 0.654 |
| | Average | 103.260* | 0.777* | 0.872* | 0.781* | 56.808* | 0.664* |

the corresponding marked area calculated for each padding case is also given. As shown in Fig. 4, the proposed sPCP and aPCP improve the performance of SRResNet. Especially, it can be seen that adaptive PCP based SRResNet has been dramatically recovering the details of textures and patterns in the marked area of the hat much more clearly than the original PCP. The generated image of the marked area in case of aPCP given in Fig. 4F is very similar to the one of the original HR sample image of Fig. 4A. In case of adaptive PCP, the MSE value corresponding to the marked area is 120.089 which is much better than the one in case of PCP, 168.955.

In *Liu et al. (2018)* and *Liu et al. (2022)*, it is shown that the performance of the original PCP is superior to other padding algorithms in various deep learning applications. So it is easily expected that sPCP and aPCP can also show superior performance in many applications of deep learning neural networks.

## PERFORMANCE ANALYSIS: U-NET FOR LUNG CT IMAGE SEGMENTATION

In this section, the performances of various padding algorithms are compared to show the general performance superiority of proposed algorithms, sPCP and aPCP. To do this, another deep learning application problem, image segmentation for lung CT images is considered, where it is important to find an accurate lung area in order to find lesions in the lung well.

As a deep learning neural network, a well-known U-Net (*Ronneberger, Fischer & Brox, 2015*) is adopted for lung image segmentation to compare the padding performance. The architectural details of the U-Net model are given below.

**Analysis path**: 1st Layer → 8 3×3 Convolutional Filters, ReLU

**Table 4  Performance comparison of padding algorithms in U-Net.**

| Padding Algorithm | Loss: Binary Crossentropy | Accuracy: Dice Coefficient | Number of Parameters | FLOPs ($\times 10^6$) | Latency (ms) |
|---|---|---|---|---|---|
| Zero Padding | 0.08224 | 0.95625 | 23,241 | 179.83 | 0.349 |
| Reflection Padding | 0.09519 | 0.95240 | 23,241 | 181.72 | 0.352 |
| Replication Padding | 0.09558 | 0.95127 | 23,241 | 181.72 | 0.354 |
| PCP | 0.07083 | 0.95756 | 23,241 | 179.83 | 0.352 |
| signed PCP (sPCP) | 0.06232 | 0.96132 | 23,241 | 179.83 | 0.546 |
| adaptive PCP (aPCP) | 0.06126 | 0.96233 | 23,241 | 179.83 | 0.580 |

activation, 1 stride, Batch Normalization, 2×2 Maxpooling

**Analysis path**: 2nd Layer → 16 3×3 Convolutional Filters, ReLU activation, 1 stride, Batch Normalization, 2×2 Maxpooling

**Analysis path**: 3rd Layer → 32 3×3 Convolutional Filters, ReLU activation, 1 stride, Batch Normalization, 2×2 Maxpooling

**Analysis path**: 4th Layer → 32 1x1 Convolutional Filters, ReLU activation, 1 stride, Batch Normalization, 2×2 Maxpooling

**Synthesis path**: 1st Layer → 2×2 Upsampling of 4th layer output of Analysis path, Concatenation with 3rd layer output of Analysis path

**Synthesis path**: 2nd Layer → 32 3×3 convolutional filters, ReLU activation, 2×2 Upsampling, Concatenation with 2nd layer output of the Analysis path

**Synthesis path**: 3rd Layer → 24 3×3 convolutional filters, ReLU activation, 2×2 Upsampling, Concatenation with 1st layer output of the Analysis path

**Synthesis path**: 4th Layer → 16 3×3 convolutional filters, ReLU activation

**Synthesis path**: 5th Layer → 64 3×3 convolutional filters, ReLU activation

**Synthesis path**: 6th Layer → Dropout @50%

**Output Layer** → 1 1x1 convolutional filter, sigmoid activation

The lung image dataset is collected in Kaggle dataset, http://www.kaggle.com/datasets/kmader/finding-lungs-in-ct-data. The dataset is consisting of 266 lung CT images and their mask ones. Among those images, 212 images are selected for training and 27 images are for validation. The remaining 27 images are used for prediction test of a trained model.

Training of U-Net continues until steady state and the best model for each padding algorithm is obtained after the training error is settled. Table 4 summarize prediction loss and accuracy as performance measures of a U-Net for training and prediction, binary crossentropy and dice similarity coefficient are taken as loss and accuracy, respectively.

Table 4 shows that the performance of the proposed algorithms is better than other existing padding ones. Latencies of the U-Net model for various padding algorithms, listed in the table, shows that sPCP and aPCP have poor latencies than PCP, as PCP calculate the ratio matrix for once and use it till end, while in sPCP and aPCP, the ratio matrix is calculated separately for each output channel and each layer.

## CONCLUSION

In this article, compensation algorithms are proposed to improve the performance of deep neural networks by efficiently handling padding issues. The performance of the suggested compensation algorithms has been analyzed and compared with the existing ones such as partial convolution based padding and zero padding. The proposed algorithms, signed PCP and adaptive PCP, show improved performance because they consider the characteristics of filter's elements.

Even though the suggested algorithms try to compensate convolutional outputs using compensation ratios simply calculated, their compensation performance is inherently limited. Therefore, it is necessary to develop an algorithm which can directly estimate exact values of padded elements in order to implement exact padding.

## FUTURE WORK

Currently, sPCP and aPCP works perfectly assuming the stride and the dilation rate equal to 1. Further is that the convolving filter shape is also considered square. These limitations are open to be addressed in the future.

### Funding

This work was supported by the Hallym University Research Fund, HRF-202208-004. The funders had no role in study design, data collection and analysis, decision to publish, or preparation of the manuscript.

### Grant Disclosures

The following grant information was disclosed by the authors:
The Hallym University Research Fund: HRF-202208-004.

### Competing Interests

The authors declare there are no competing interests.

### Author Contributions

- Safi Ullah conceived and designed the experiments, performed the experiments, analyzed the data, performed the computation work, prepared figures and/or tables, and approved the final draft.
- Seong-Ho Song conceived and designed the experiments, authored or reviewed drafts of the article, and approved the final draft.

### Data Availability

The data is available at Kaggle: www.kaggle.com/datasets/kmader/finding-lungs-in-ct-data.

## Supplemental Information

Supplemental information for this article can be found online at http://dx.doi.org/10.7717/peerj-cs.2287#supplemental-information.

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
