# Peer review of "Design of compensation algorithms for zero padding and its application to a patch based deep neural network"

_PeerJ Computer Science, doi:10.7717/peerj-cs.2287_

## Round 0.1 · original submission · Major Revisions

I reviewed the paper before sending it out to other reviewers. I understand what exactly you are proposing. However, several issues must be addressed before I would like to send it to the reviewers.

1. The language. Please have someone read and fix several things, including clarity in presentation, communication of the concept clearly, and sentence structure in many places. I do not want reviewers to get stuck on language and not pay attention to the technical content.

2. Present your reasoning of why the two schemes develop make sense intuitively and then present the details. Right now, it read that here are two ways we tried and they seem to work in the examples we worked on.

3. Present it algorithmically on how and when to apply your proposed scheme systematically. The paper should communicate that you are presenting a method/algorithm as opposed to summarizing a work you did in some context.

If you can accomplish them, I will be happy to have this reviewed externally. Right now, even as a reviewer, I will recommend to reject it.

Thanks for your interest in the journal.

---

## Round 0.2 · Minor Revisions

Thanks for your interest in the journal and your contributions to the journal's mission. Please pay careful attention to the reviewers' comments and suggestions, in particular including more experimental results.

Reviewer 1 ·

Basic reporting

1) Only one application (Super Resolution) and one model (SRResNet) are tested, and a handful of datasets are used for evaluation. The proposed methodology is not application-dependent. It is good to show the effectiveness in different domains. It's good to see the results for a different application like denoising and models which require zero padding in their network.

2) It will be useful to report the latency or throughput on the hardware along with FLOPs, number of parameters and GPU memory. A technique which requires fewer FLOPs may not require less latency. Also, the proposed methodology should be compared with the earlier works not only on the validation datasets but also on the hardware performance aspect to observe the tradeoff.

3) A paper should be able to read even if we remove all the references. A reference should not be used directly. For example, in Line 63, the authors have written “Cube padding in (Cheng et al. 2018) suggested base” Instead, it should be “Cube padding by Cheng et al. (Cheng et al. 2018) suggested base.”

Experimental design

--

Validity of the findings

--

Reviewer 2 ·

Basic reporting

Authors proposed sPCP and aPCP as a compensation for zero padding to enhance the performance of deep CNN. These algorithms show improved performance with inherent limits.

Experimental design

In line 49, authors mentioned that in SR tasks, DPP are more sensitive to zero padding and can be deteriorated by it.

We suggest authors to extend the experiment to multiple image tasks including classification and segmentation to comprehensively validate the findings.

Also, we encourage authors to introduce more classic padding algorithms for comparison such as reflection padding, mirror padding, circular padding, etc. They can be implemented with little effort.

Validity of the findings

We find this work is somewhat similar to a previous work of the authors, titled "SRResNet Performance Enhancement Using Patch Inputs and Partial Convolution-Based Padding". Some key figures are identical.

We hope authors provide new figures in this submission.

---

## Round 0.3 · accepted · Accept

Thanks for responding to the reviewers' comments and being responsive to a great extent. I am now ready to recommend it for publication.